# Genetic Analysis Reveals a Protective Effect of Sphingomyelin on Cholelithiasis

**DOI:** 10.3390/genes16050523

**Published:** 2025-04-29

**Authors:** Kun Mao, Ang Li, Haochen Liu, Yuntong Gao, Ziyan Wang, Xisu Wang, Shixuan Liu, Ziyuan Gao, Jiaqi Quan, Moyan Shao, Yunxi Liu, Liang Shi, Bo Zhang, Tianxiao Zhang

**Affiliations:** 1Department of Epidemiology and Biostatistics, School of Public Health, Xi’an Jiaotong University Health Science Center, Xi’an 710061, China; submacar@stu.xjtu.edu.cn (K.M.); 2242211608@stu.xjtu.edu.cn (A.L.); lhckrst@stu.xjtu.edu.cn (H.L.); altogik2002@stu.xjtu.edu.cn (Y.G.); yanww0502@163.com (Z.W.); wxs_doofzijing@stu.xjtu.edu.cn (X.W.); liushixuan2022@gmail.com (S.L.); gzy168@stu.xjtu.edu.cn (Z.G.); qmzpal031021@163.com (J.Q.); 15518778156@163.com (M.S.); 1163056478@stu.xjtu.edu.cn (Y.L.); shi-liang@stu.xjtu.edu.cn (L.S.); 2Xi’an Jiaotong University Health Science Center, Xi’an 710061, China; 3School of Life Science and Technology, Xi’an Jiaotong University, Xi’an 710049, China; 4The Key Laboratory of Biomedical Information Engineering of Ministry of Education, Xi’an Jiaotong University, Xi’an 710049, China; 5The School of Mathematics and Statistics, Xi’an Jiaotong University, Xi’an 710049, China; 6Department of Geriatric Endocrinology Metabolism, The Second Affiliated Hospital of Xi’an Jiaotong University, Xi’an 710004, China

**Keywords:** Mendelian randomization, choline metabolites, cholelithiasis, low-density lipoprotein, colocalization analysis

## Abstract

Background: Cholelithiasis is the most common disorder affecting the biliary system. Choline is an essential nutrient in the human diet and is crucial for the synthesis of neurotransmitters. Previous studies have suggested an association between choline metabolites and cholelithiasis. However, the underlying mechanisms remain unclear. This research aims to fill the knowledge gap regarding the role of choline metabolites in cholelithiasis. Methods: Genetic data related to choline metabolites and other covariates were retrieved from the U.K. Biobank and IEU OpenGWAS database. Two-sample (TSMR) and multivariate Mendelian randomization (MVMR) analyses, mediation analysis, linkage disequilibrium score regression (LDSC), colocalization analysis, and enrichment analysis were performed. Results: A significant causal relationship was identified between serum level of sphingomyelin and cholelithiasis (*p*-value = 0.0002). A protective causal effect was identified in MVMR analysis. The following mediated MR analysis indicated that only LDL mediated a large part of the causal relationship (59.18%). Seven genes, including *GCKR*, *SNX17*, *ABCG8*, *MARCH8*, *FUT2*, *APOH*, and *HNF1A*, were revealed to be colocalized with the causal signal between sphingomyelin and cholelithiasis. Conclusion: The present study has identified a protective effect between sphingomyelin and cholelithiasis. This effect is largely mediated by LDL. The findings of this study offer valuable information for further exploration of the molecular mechanisms of cholelithiasis.

## 1. Introduction

Cholelithiasis is a common condition characterized by the accumulation of elevated levels of cholesterol or bilirubin—a breakdown product of hemoglobin—in the bile, which leads to the formation of stones within the biliary system, including the gallbladder and bile ducts. Approximately 20% of adults are affected by cholelithiasis, with more than 20% of these individuals developing symptoms and complications. Consequently, cholelithiasis imposes a substantial socioeconomic burden as a prevalent and costly digestive disorder [1]. Extensive research has identified several risk factors for cholelithiasis. Non-modifiable factors, including race, female sex, pregnancy, and age over 40 years, play a particularly significant role. Modifiable risk factors for cholelithiasis include obesity, a high-calorie diet, metabolic syndrome, and dyslipidemia. At present, surgical intervention constitutes the primary treatment for symptomatic cholelithiasis [2].

Choline, an essential nutrient in the human diet, is crucial for the synthesis of neurotransmitters such as acetylcholine, as well as for the production of methyl donors, betaine, and phospholipids. It plays vital roles in cell maintenance and growth throughout life, including neurotransmission, membrane synthesis, lipid transport, and single-carbon metabolism [3]. It is imperative to note that essential choline exists in several forms, including water-soluble free choline, fat-soluble phosphatidylcholine, and sphingomyelin. These forms are absorbed and metabolized through distinct pathways, which influence their bioavailability [4]. Previous studies have suggested an association between sphingolipids and cholelithiasis [5]. However, the underlying mechanisms between choline metabolites and cholelithiasis remain unclear.

Mendelian randomization (MR) is a highly efficacious tool for the identification of causal risk factors that underpin complex traits and diseases by means of genetic data. By employing single-nucleotide polymorphisms (SNPs) from genome-wide association studies (GWAS) as instrumental variables (IVs) for exposure, MR controls confounding factors and avoids reverse causation, strengthening causal inferences [6,7,8,9]. In the present study, the causal relationship between choline metabolite levels and cholelithiasis was examined using MR analysis. This current research aims to fill the knowledge gap regarding the role of choline metabolites in cholelithiasis and provide valuable insights for developing more effective preventive and treatment strategies. The graphical abstract is presented in Appendix A.

## 2. Materials and Methods

### 2.1. Study Design

The study design and the assumptions of an MR study are presented in Figure 1. The risk factors under consideration include choline metabolites and other covariates, including LDL, HDL, triglycerides, and coronary artery disease (CAD), which were retrieved from specific data sources. The analyses include MR, LDSC, colocalization analysis, and enrichment analysis. The selection of genetic markers adheres to the three fundamental principles of instrumental variables. Firstly, the genetic markers must demonstrate a strong association with the exposure. Secondly, they must not demonstrate a direct association with the outcome. Thirdly, they must influence the outcome exclusively through their effect on the exposure. This study was conducted in strict accordance with the STROBE-MR checklist [10]. The code employed in this article is available in Appendix A.

### 2.2. Data Sources

The serum levels of three choline metabolites—total choline, phosphatidylcholine, and sphingomyelin—were selected as exposure factors from the U.K. Biobank (n = 114,999). The GWAS summary statistics for these metabolites are accessible via the IEU OpenGWAS database (https://gwas.mrcieu.ac.uk, accessed on 10 November 2024), with identifiers of met-d-cholines, met-d-phosphatidylcholines, and met-d-sphingomyelins (Borges CM) [11]. The genetic data of cholelithiasis from the Neale Lab within the U.K. Biobank were used for endpoints (http://www.nealelab.is/uk-biobank, accessed on 10 November 2024). The IEU OpenGWAS database (GWAS identifier: ukb-a-559, n = 337,199) was utilized for the purpose of data retrieval, and cholelithiasis was diagnosed according to ICD-10 (K80). For MVMR, four additional covariates were selected and adjusted in the analyses: LDL (ieu-b-5089, n = 201,678), HDL (ieu-b-109, n = 403,943), triglycerides (ieu-b-111, n = 441,016) [12], and CAD (ebi-a-GCST90013868, n = 352,063) [13]. The serum levels of three choline metabolites—total choline, phosphatidylcholine, and sphingomyelin—were selected as exposure factors from the U.K. Biobank (n = 114,999). The GWAS summary statistics for these metabolites are accessible via the IEU OpenGWAS database (https://gwas.mrcieu.ac.uk, accessed on 10 November 2024), with identifiers of met-d-cholines, met-d-phosphatidylcholines, and met-d-sphingomyelins (Borges CM) [11]. The genetic data of cholelithiasis from the Neale Lab within the U.K. Biobank were used for endpoints (http://www.nealelab.is/uk-biobank, accessed on 10 November 2024). The IEU OpenGWAS database (GWAS identifier: ukb-a-559, n = 337,199) was utilized for the purpose of data retrieval, and the diagnosis of cholelithiasis was made in accordance with the International Classification of Diseases, Tenth Revision (ICD-10, K80). In the context of multivariate Mendelian randomization (MVMR), the analysis incorporated four additional covariates, which were selected and adjusted as follows: LDL (ieu-b-5089, n = 201,678), HDL (ieu-b-109, n = 403,943), triglycerides (ieu-b-111, n = 441,016) [12], and CAD (ebi-a-GCST90013868, n = 352,063) [13]. The genetic data were derived exclusively from European populations and had undergone rigorous quality control measures and ethical approval, thus ensuring direct applicability.

### 2.3. Genetic Instrument Selection

SNPs with strong correlations (*p*-value < 5 × 10^−8^) were initially selected as candidate IVs. Linkage disequilibrium (LD) among them was estimated. One genetic marker was excluded on the basis that the r^2^ values of the corresponding SNP pair were greater than 0.001 (with a distance of less than 10,000 kb). To remove weak instrumental variables, an F-test was performed for each SNP, and SNPs with F-values less than 10 were excluded. Phenoscaner (https://github.com/phenoscanner/phenoscanner, accessed on 10 November 2024) was utilized to search and remove confounding SNPs [14,15]. Finally, a total of 49, 48, and 48 SNPs were identified as IVs for total choline, phosphatidylcholine, and sphingomyelin, respectively.

### 2.4. TSMR and MVMR Analyses

Mendelian randomization satisfies the relevance assumption, the independence assumption, and the exclusion restriction [16]. Five MR methods, including MR-Egger [17], weighted median [18], inverse variance weighted (IVW) [19], simple mode, and weighted mode [20], were utilized for MR analyses using the TwoSampleMR package (http://www.r-project.org, accessed on 10 November 2024). The IVW estimates were selected as the primary approach, while the other four methods were employed to enhance robustness across diverse scenarios. A significant causal relationship was considered when the *p*-value of IVW was less than 0.05, and the results of all five methods had the same direction. Heterogeneity was assessed using the MR-Egger Cochran Q test [21]. The MR-Egger intercept [22] and the MR-Presso global test [23] were utilized to test for pleiotropy. In instances where heterogeneity and pleiotropy were detected, the MR-Presso outlier method [23] was employed to remove outliers, and the analysis was repeated. Furthermore, leave-one-out tests for each SNP were conducted using the IVW results [24], and the SNP effect values were visualized in a funnel plot [25] to ensure that no outliers were interfering with the model results.

MVMR was carried out including LDL, HDL, triglycerides, and CAD as covariates. Results were evaluated using the IVW random effects model and the MR-Egger model.

### 2.5. Reverse and Mediated Mendelian Randomization Analysis

Reverse and mediated MR analysis was conducted for sphingomyelin, which was identified as a significant positive causal factor for cholelithiasis. The analysis incorporated four factors as mediators. In the analysis of the mediator Mendelian randomization, the effect of sphingomyelin on LDL was β1, the effects of the mediators on cholelithiasis were β2, and the indirect effect was calculated using β1 × β2. The significance of the indirect effect was then tested using a stepwise testing method.

### 2.6. LDSC and Colocalization Analysis

LDSC and colocalization analyses were performed to investigate genetic correlations between the three exposure factors and cholelithiasis [26,27,28]. The coloc package was used for colocalization analysis (*p*1 = 1 × 10^−4^, *p*2 = 1 × 10^−4^, *p*12 = 1 × 10^−5^) [28]. SNPs associated with exposure factors were selected to define the colocalization regions based on a significance threshold of *p*-value < 1 × 10^−6^. Linkage disequilibrium (LD) pruning was performed within a 100 kb window by removing SNPs with r^2^ < 0.001. For the remaining SNPs, regions of colocalization were defined by extending 50 kb on either side of the selected SNPs. Subsequently, a full Bayesian colocalization analysis was conducted, utilizing Bayes factors. The threshold for posterior probability for hypothesis 4 (PPH4) was set at > 0.9.

### 2.7. Gene Enrichment Analysis

Gene enrichment analysis was further conducted for loci obtained from colocalization analysis using the NCBI website (https://www.ncbi.nlm.nih.gov/, accessed on 10 November 2024) and the STRING database (https://string-db.org/, accessed on 10 November 2024) [29]. Enrichment analysis was then performed using GO and KEGG databases [30].

## 3. Results

### 3.1. Significant Signals Were Identified for Causal Relationship Between Sphingomyelin and Cholelithiasis

In TSMR analysis, heterogeneity was detected (*p*-value < 0.05). The MR-Presso method excluded specific SNPs for total choline, phospholipids, and sphingolipids, respectively. Subsequently, a significant causal relationship was identified between serum level of sphingomyelin and cholelithiasis (OR [95%CI] = 1.0049 [1.0028–1.0070], *p*-value = 0.0002) (Figure 2). No horizontal pleiotropy was detected by MR-Egger intercepts, while the MR-Presso global test indicated some presence (Appendix A). Leave-one-out tests and funnel plots did not identify SNPs with a major impact on the results (Appendix A).

### 3.2. Low-Density Lipoprotein (LDL) and High-Density Lipoprotein (HDL) Might Mediate Part of the Causal Relationship Between Sphingomyelin and Cholelithiasis

A protective causal effect was identified in the MVMR analysis (Table 1). The following mediated MR analysis indicated that only LDL and HDL mediated part of the causal relationship between sphingomyelin and cholelithiasis (Figure 3, Appendix A). The indirect effects of LDL were found to be −0.0029, and the indirect effects of HDL were −0.0018. The LDL accounted for 59.18% of the total effect of sphingomyelin on cholelithiasis; however, the mediated proportion of HDL did not reach statistical significance. Reverse MR analysis revealed that the causal relationship between cholelithiasis and sphingolipids was not significant (*p*-value for IVW = 0.2520, *p*-value for MR-Egger intercept = 0.1790, *p*-value for Cochran Q test = 3.7161 × 10^−28^).

### 3.3. Bioinformatics Evidence for Unraveling the Significant Causal Signal Between Sphingomyelin and Cholelithiasis

LDSC analysis demonstrated genetic correlations between serum levels of three risk factors (total choline, phosphatidylcholine, sphingomyelin) and cholelithiasis at the whole-genome level (Appendix A). Colocalization analysis revealed abundant colocalization signals, thereby identifying specific SNPs mapped to genes, including GCKR (glucokinase regulatory protein), SNX17 (sorting nexin-17), ABCG8 (ATP-binding cassette sub-family G member 5), MARCH8 (membrane-associated ring-CH-type finger 8), FUT2 (fucosyltransferase 2), APOH (apolipoprotein H) and HNF1A (hepatocyte nuclear factor 1α) (Figure 4, Appendix A).

These seven genes were found to be enriched in multiple biological processes, cellular components, and molecular functions in the GO database. In the KEGG database, these genes were identified to be enriched in glycosphingolipid biosynthesis—lacto and neolacto series, cholesterol metabolism, maturity-onset diabetes of the young, glycosaminoglycan biosynthesis—keratan sulfate, and glycosphingolipid biosynthesis—globo and isoglobo series (Appendix A).

## 4. Discussion

Choline, a trace element in plasma that performs crucial physiological functions in the human body, is often overlooked in the development of disease. In the present study, a significant causal relationship was identified between sphingomyelin, important choline metabolites, and cholelithiasis. Further analysis suggested that this protective causal effect may be mediated by serum levels of HDL and LDL. In addition, seven genes enriched in glycosphingolipid biosynthesis were also identified as contributing to sphingomyelin and cholelithiasis through colocalization analysis.

TSMR analysis revealed a significant causal relationship between sphingomyelin and cholelithiasis after the elimination of outliers. MVMR analysis was used to adjust for suspected confounders and account for pleiotropy. A protective effect was identified through MVMR between serum levels of sphingomyelin and cholelithiasis. These findings were inconsistent with a previous study performed by Jiarui et al. [5] This finding was further illustrated in followed mediating analysis. LDL and HDL levels play an important mediating role between sphingolipids and cholelithiasis. TSMR showed that sphingolipids directly increased the risk of cholelithiasis but also increased LDL. The indirect effect of sphingolipids in reducing the risk of cholelithiasis by raising LDL was significantly greater than their direct effect, as shown by the results of MVMR. Thus, the overall effect of sphingolipids is protective against cholelithiasis. The effect of HDL was similar to that of LDL, but its effect was smaller. This partly explains why adjusting for HDL in MVMR did not change the causal relationship between sphingolipids and cholelithiasis.

LDSC analysis identified genome-wide genetic correlations between three choline metabolites and cholelithiasis. Subsequent colocalization analysis revealed seven potential colocalized loci between sphingomyelin and cholelithiasis, suggesting that multiple genes may contribute to this genetic correlation. It provides a brief discussion of the functions of the seven identified genes that would help elucidate their potential roles in the genetic correlation between sphingomyelin and cholelithiasis. Among the colocalized genes, *GCKR* functions in the liver through protein–protein interactions, facilitating its nuclear localization [31]. *HNF1A* is involved in glycolipid metabolism [32]. ABCG5 encodes a transporter protein involved in the excretion of cholesterol and promotes cholesterol excretion [33]. Common mutations in ABCG5 confer the majority of the genetic risk for cholelithiasis, accounting for approximately 25% of the total risk [1]. Previous studies have shown that patients with cholesterol stone disease and cholecystitis have increased levels of ABCG5 expression [34]. In conclusion, choline metabolites may combat cholelithiasis by promoting cholesterol excretion and transport. Multiple lines of evidence indicate that these three genes are associated with diabetes, as reported in previous studies [35,36,37]. However, the present study is constrained by its reliance on public databases; further experimental validation is required to enhance its credibility. Subsequent studies may contribute to the enhancement of the study’s credibility. Cholelithiasis is also a prevalent gastrointestinal manifestation of diabetes, suggesting a genetic correlation between the two conditions [38]. Sphingomyelin has been identified to be associated with diabetes [39] and has also been demonstrated to influence the progression of diabetic kidney disease [40]. Enrichment analysis also demonstrated that the localized genes are implicated in lipid and carbohydrate metabolism and are associated with maturity-onset diabetes in young people. These findings suggest that the underlying molecular mechanisms responsible for sphingomyelin levels may contribute to the development of both cholelithiasis and diabetes. This study suffers from several limitations. Firstly, the analysis was restricted to European ancestry samples, which limits the generalisability of the findings. Further validation studies are still needed to generalize the findings to other populations. Additionally, due to limitations in the data availability, not all choline metabolites were analyzed. In order to obtain more robust results, larger data sets from multiple sources are required. While the study provides insights into the role of choline metabolites in cholelithiasis etiology, replication in more diverse populations is needed for definitive conclusions to be drawn. Furthermore, the current research reveals causal reference, and would not have great significance for clinical practice. Related clinical translational research is needed in the future.

## 5. Conclusions

In conclusion, the present study has identified a protective effect between sphingomyelin and cholelithiasis. This effect is largely mediated by LDL. The findings of this study offer valuable information for further exploration of the molecular mechanisms of cholelithiasis.

## Figures and Tables

**Figure 1 genes-16-00523-f001:**
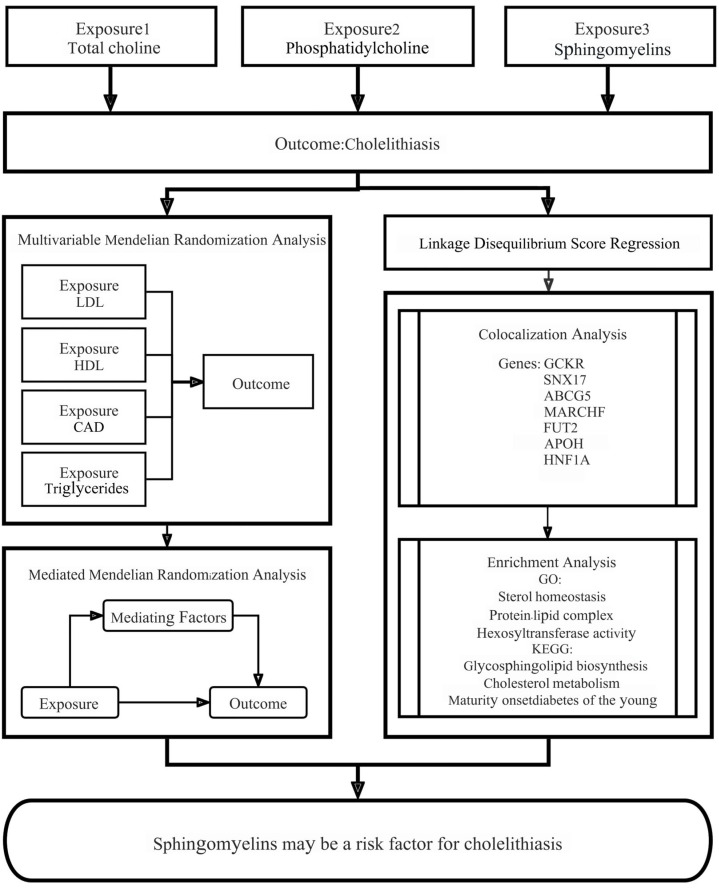
Conceptual framework of this study.

**Figure 2 genes-16-00523-f002:**
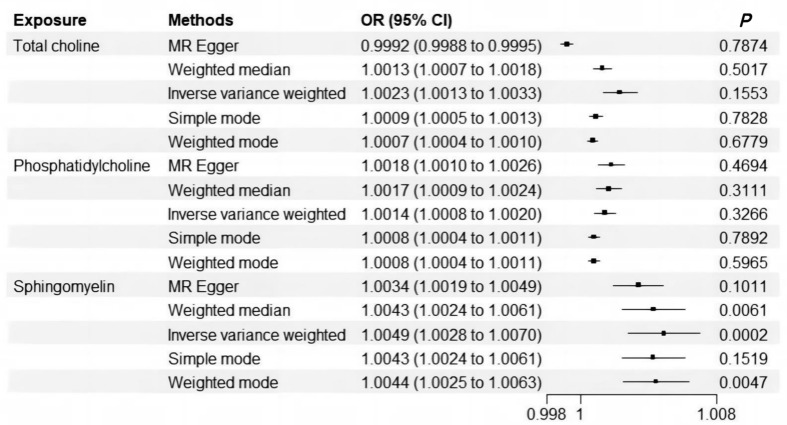
Mendelian randomization analysis of the relationship between choline metabolites and cholelithiasis (results corresponding to five different methods).

**Figure 3 genes-16-00523-f003:**
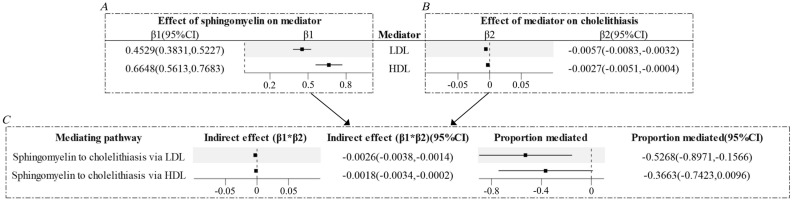
Forest plots of mediated Mendelian randomization analysis. (**A**) MR-estimated effects of sphingomyelin on LDL and HDL, presented as β1 with 95% CI. (**B**) MR-estimated effects of LDL and HDL on cholelithiasis, presented as β2 with 95% CI. (**C**) Indirect effects and proportions mediated of each mediator separately, by product of coefficients method with delta method estimated 95% CIs.

**Figure 4 genes-16-00523-f004:**
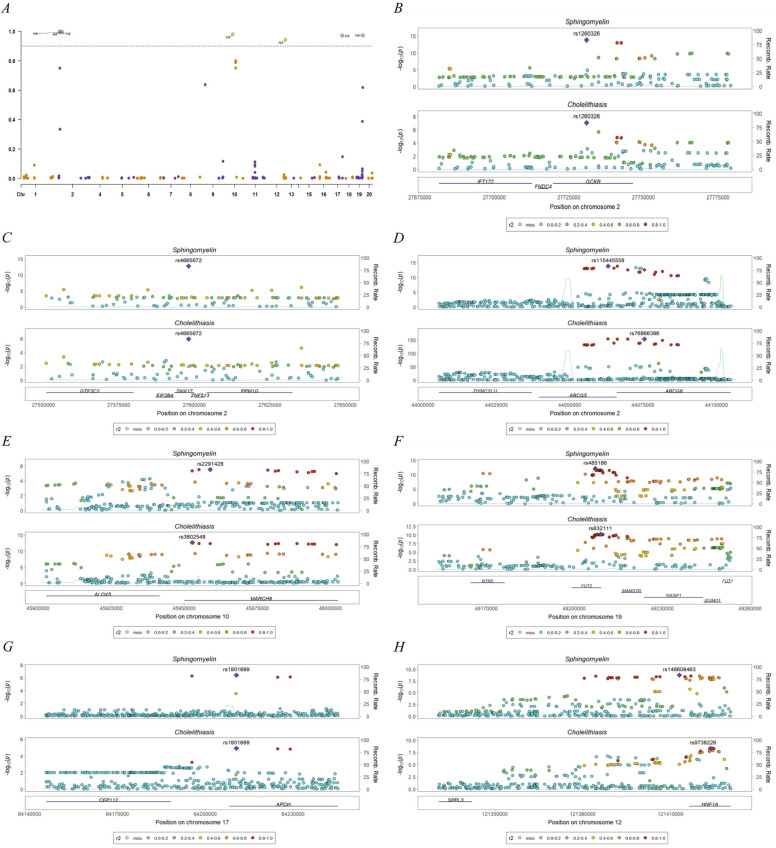
Manhattan plots for colocalization analysis. (**A**) Manhattan plot of selected SNPs associations with PPH4 at the genome-wide scale, the vertical coordinate indicates PPH4. Yellow and purple dots are used to distinguish between different chromosomes. (**B**–**H**) Locus comparison plot for the colocalization regions (PPH4 > 0.9) that were identified through the implementation of the COLOC analysis. The purple dot is used to denote the independent SNPs associated with genetic liability. The corresponding “selected SNPs” are rs1260326, rs4665972, 1s56266464, 1s3802548, 1s35106244, 1575003668 and rs2649999 in order.

**Table 1 genes-16-00523-t001:** Multivariable Mendelian randomization analysis between choline metabolites and cholelithiasis.

Outcome	Exposure	nSNPs	MVMR-IVW		MVMR-Egger		*p* for MR-Egger Intercept	F-Value
			OR (95% CI)	*p*-Value	OR (95% CI)	*p*-Value		
Cholelithiasis	HDL	315	1.0000 (1.0000 to 1.0000)	0.9990	1.0052 (1.0029 to 1.0074)	0.0122	0.0002	17.6444
	Total choline		0.9961 (0.9944 to 0.9978)	0.0590	0.9956 (0.9937 to 0.9975)	0.0308		11.2519
	LDL	90	0.9931 (0.9901 to 0.9961)	0.0671	0.9932 (0.9903 to 0.9962)	0.1200	0.9559	52.8175
	Total choline		0.9977 (0.9967 to 0.9987)	0.5898	0.9978 (0.9969 to 0.9988)	0.6240		47.1951
	Triglyceride	275	1.0002 (1.0001 to 1.0002)	0.8990	0.9964 (0.9948 to 0.9979)	0.0450	0.0039	116.1858
	Total choline		0.9984 (0.9978 to 0.9991)	0.3330	0.9982 (0.9974 to 0.9990)	0.2580		22.2594
	CAD	84	0.9990 (0.9985 to 0.9994)	0.4017	0.9976 (0.9965 to 0.9986)	0.1762	0.2870	30.1093
	Total choline		0.9956 (0.9936 to 0.9975)	0.0083	0.9951 (0.9930 to 0.9972)	0.0047		78.4745
	HDL	316	0.9994 (0.9991 to 0.9996)	0.6820	1.0043 (1.0024 to 1.0061)	0.0388	0.0004	20.3171
	Phosphatidylcholine		0.9976 (0.9965 to 0.9986)	0.2170	0.9971 (0.9958 to 0.9984)	0.1313		12.9910
	LDL	91	0.9926 (0.9894 to 0.9958)	0.0404	0.9924 (0.9891 to 0.9957)	0.0779	0.9270	68.9316
	Phosphatidylcholine		0.9989 (0.9985 to 0.9994)	0.7796	0.9988 (0.9983 to 0.9993)	0.7696		58.3947
	Triglyceride	277	1.0004 (1.0002 to 1.0005)	0.7690	0.9964 (0.9949 to 0.9980)	0.0502	0.0027	108.2092
	Phosphatidylcholine		0.9986 (0.9980 to 0.9992)	0.3500	0.9984 (0.9977 to 0.9991)	0.2956		25.0179
	CAD	84	0.9988 (0.9982 to 0.9993)	0.3477	0.9973 (0.9962 to 0.9985)	0.1669	0.3105	29.6764
	Phosphatidylcholine		0.9971 (0.9958 to 0.9984)	0.0868	0.9967 (0.9952 to 0.9981)	0.0577		88.3187
	HDL	309	1.0028 (1.0016 to 1.0040)	0.3119	1.0096 (1.0054 to 1.0137)	0.0051	0.0009	9.2775
	Sphingomyelin		0.9907 (0.9866 to 0.9947)	0.0146	0.9901 (0.9858 to 0.9944)	0.0089		6.8220
	LDL	90	0.9945 (0.9921 to 0.9969)	0.2890	0.9947 (0.9924 to 0.9970)	0.3420	0.9368	23.9513
	Sphingomyelin		0.9969 (0.9956 to 0.9983)	0.5530	0.9970 (0.9957 to 0.9983)	0.5710		24.7277
	Triglyceride	273	0.9989 (0.9984 to 0.9994)	0.5529	0.9953 (0.9933 to 0.9973)	0.0972	0.0895	66.8256
	Sphingomyelin		0.9943 (0.9919 to 0.9968)	0.0297	0.9940 (0.9914 to 0.9966)	0.0212		19.0968
	CAD	90	0.9982 (0.9974 to 0.9990)	0.4367	0.9983 (0.9976 to 0.9991)	0.6246	0.9627	28.3273
	Sphingomyelin		0.9923 (0.9889 to 0.9957)	0.0231	0.9923 (0.9890 to 0.9957)	0.0273		67.9384

MVMR-Egger: multivariable Mendelian randomization using Egger regression; MVMR-IVW: multivariable Mendelian randomization using inverse variance-weighted approach; nSNPs: number of SNPs used in MR; LDL: low-density lipoprotein; HDL: high-density lipoprotein; CAD: coronary artery disease.

## Data Availability

The original data presented in the study are openly available. The summarized GWAS data were extracted from IEU OpenGWAS (https://gwas.mrcieu.ac.uk/, accessed on 10 November 2024), U.K. Biobank (http://www.nealelab.is/uk-biobank, accessed on 10 November 2024), and GWAS catalog (https://www.ebi.ac.uk/gwas/, accessed on 10 November 2024).

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
