# Peer review of "Genetic Analysis Reveals a Protective Effect of Sphingomyelin on Cholelithiasis"

_genes, 2025, doi:10.3390/genes16050523_

Round 1
Reviewer 1 Report
Comments and Suggestions for Authors
The manuscript presents a comprehensive Mendelian randomization analysis to explore the causal relationship between choline metabolites, particularly sphingomyelin, and cholelithiasis. However, there are some points that I recommend addressing to further strengthen the manuscript:
1. In several instances (e.g., lines 144–148), the wording implies that higher sphingomyelin levels increase risk, which contradicts the reported protective effect. Please clarify the direction of the causal relationship consistently throughout the manuscript.
2. In Line 69, references [7–9] pertain to LDSC and COLOC analyses, not Mendelian randomization. It is recommended to replace them with more appropriate citations, such as PMID: 39003418, PMID: 38333712, and PMID: 37537391.
3. Data and code need to be shared either through a code-sharing repo like GitHub or a docker-like system such as codeocean for clear reproducibility of the work.
4. Author should use the STROBE-MR checklist to improve the reporting of MR studies and cite PMID: 37198682.
5. Author should further check key expression in tissues. This manuscript consists solely of bioinformatics analysis of public databases which are not accompanied by validation (independent cohort or biological validation in vitro or in vivo).
6. A conclusion figure (graphical abstract) will be very useful for the readers.
7. Some effect sizes (e.g., OR = 1.0049) are very close to 1. While statistically significant, the clinical relevance should be discussed.
8. There are minor grammatical errors throughout the text. For instance: Line 169: "This study is suffered from…" → revise to "This study suffers from…".
Reviewer 2 Report
Comments and Suggestions for Authors
This manuscript presents a well-structured and scientifically rigorous investigation into the causal role of choline metabolites—especially sphingomyelin—in the development of cholelithiasis, using comprehensive Mendelian randomization methodologies. The study is methodologically sound, integrates multi-layered genomic analyses (TSMR, MVMR, mediation, colocalization), and yields biologically meaningful results that have implications for both metabolic research and gallstone disease prevention strategies.
Major Concerns
-
Similarity Index and Language Reuse
As noted above, the manuscript’s 31% similarity score is relatively high. While some overlap is expected in methods-heavy studies, the authors are strongly encouraged to rephrase sections that may rely too closely on previously published language, even if self-authored. This will enhance both the scientific and editorial integrity of the submission. -
Conceptual Clarity and Terminology
The manuscript refers to sphingomyelin as both a "risk factor" and "protective" agent at different points (e.g., lines 137 vs. 147), which creates confusion. This contradiction should be resolved to preserve conceptual clarity. -
Population-Specific Findings
The authors correctly acknowledge that the analyses are limited to individuals of European ancestry. This limitation should be more explicitly emphasized when discussing potential generalizability, especially in the Conclusions section.
Minor Comments-
Figure Legends and Supplementary Material
Ensure that figures (especially Figures 2 and 3) are accompanied by self-explanatory legends. References to supplementary figures and tables in the main text could be more consistent and visible. -
English Language Editing
Several passages, particularly in the Discussion, would benefit from minor stylistic and grammatical improvements. A language polishing service or native speaker review is recommended. -
Mechanistic Interpretation
While the statistical mediation via LDL is well described, a brief mechanistic hypothesis linking sphingomyelin, lipoprotein metabolism, and gallstone formation would strengthen the biological narrative. - The iThenticate similarity report indicates a 31% similarity index. While this does not necessarily imply plagiarism, it may point to excessive reuse of language from prior publications or methodological sections. The authors are encouraged to review and revise the manuscript carefully, especially the Introduction and Methods sections, to improve originality of expression while maintaining scientific accuracy.
-
Round 2
Reviewer 1 Report
Comments and Suggestions for Authors
well revision